# Kisspeptin-10 Ameliorates Obesity-Diabetes with Diverse Effects on Ileal Enteroendocrine Cells and Pancreatic Islet Morphology in High-Fat Fed Female Mice

**DOI:** 10.3390/biom15111591

**Published:** 2025-11-13

**Authors:** Ananyaa Sridhar, Dawood Khan, Rithiga Muthukumar, Swetha Sampathkumar, Nigel Irwin, Peter R. Flatt, R. Charlotte Moffett

**Affiliations:** Diabetes Research Centre, Biomedical Sciences Research Institute, School of Pharmacy and Pharmaceutical Sciences, Ulster University, Coleraine BT52 1SA, UK; a.sridhar@ulster.ac.uk (A.S.); d.khan@ulster.ac.uk (D.K.); pr.flatt@ulster.ac.uk (P.R.F.); cr.moffett@ulster.ac.uk (R.C.M.)

**Keywords:** kisspeptin, high-fat diet, islet, enteroendocrine cell, morphology

## Abstract

Kisspeptin is a neuropeptide recognised for a pivotal role within the reproductive system, but potentially important endocrine metabolic effects are less well understood. We examined effects of twice-daily intraperitoneal administration of saline vehicle or kisspeptin-10 (25 nmol/kg), for 21 days, on glucose homeostasis, energy balance, circulating hormones as well as the morphology-function of enteroendocrine and islet cells in high-fat diet (HFD) fed female mice, with normal diet (ND) mice as an additional control group. Kisspeptin-10 decreased body weight, blood glucose and energy intake to ND levels. HFD increased circulating follicle-stimulating hormone (FSH) levels, which were further enhanced by kisspeptin-10 along with luteinising hormone (LH) concentrations. Neither HFD nor kisspeptin-10 affected progesterone or corticosterone. In the ileum, kisspeptin-10 decreased crypt depth and restored villi length to ND control levels, as well as increasing the proportion of glucose-dependent insulinotropic polypeptide (GIP) positive cells when compared to HFD mice and glucagon-like peptide-1 (GLP-1) positive cells compared to ND mice. Peptide YY (PYY) immunoreactivity was unaltered by HFD or kisspeptin-10. Plasma GIP was unchanged but circulating GLP-1 and PYY were reduced to ND levels. Within the pancreas, total islet, beta- and alpha-cell areas were similar in all mice, but kisspeptin-10 intervention restored relative insulin area to ND levels. Glucagon radius, an indicator of peripherally located alpha-cells, was reduced in HFD mice but normalised by kisspeptin-10 alongside elevated glucagon-islet area. Notably, beta-cell proliferation was increased by kisspeptin-10 with no alteration in beta-cell apoptosis. Overall, we reveal a previously uncharacterised diverse metabolic role for kisspeptin in directly modulating the gut–pancreatic axis.

## 1. Introduction

Pulsatile secretions of regulatory hormones influence both metabolic and reproductive physiology, especially in females, implying a connection between energy homeostasis and reproductive function [1]. However, obesity can disrupt this balance by negatively altering secretion and/or action of these hormones, leading to a cascade of endocrine disturbances that impair fertility and metabolism [2]. In brief, chronic nutritional excess alters gonadal steroidogenesis, peripheral insulin sensitivity and overall energy regulation [3]. Such dietary-induced disturbances reflect a complex interplay between central neuroendocrine circuits and peripheral metabolically active tissues.

In this context, kisspeptin, a 54-amino acid neuropeptide, initially characterised as a metastasis-suppressor gene, and then subsequently identified as a regulator of the hypothalamic–pituitary–gonadal (HPG) axis and reproductive function [4], exerts effects in obesity-related metabolic dysfunction that still need to be fully elucidated. In keeping with this, kisspeptin and its receptor, KISS1R or GPR54, are expressed both centrally and in metabolically active tissues, including the pancreas and gastrointestinal tract, supporting broader physiological relevance beyond reproductive control [5]. Indeed, accumulating evidence demonstrates that kisspeptin influences insulin secretion, glucose homeostasis and appetite, possibly through interactions with hypothalamic neurocircuits [6]. Further to this, in humans, peripheral administration of kisspeptin enhances glucose-stimulated insulin secretion [7], with similar observations made in isolated murine islets [8], highlighting a direct role of kisspeptin on pancreatic beta-cell function. That said, there are also reports of inhibitory effects of kisspeptin on insulin secretion [9,10,11], suggesting that it is multifaceted.

Beyond the endocrine pancreas, GPR54 signalling is also implicated in the maintenance of normal gut physiology and motility [12]. Moreover, there is now a well-established gut–pancreas communication axis that is vital for the efficient regulation of metabolism [13], with diet known to directly influence this reciprocal relationship [14]. Thus, in addition to inducing compensatory beta-cell proliferation and hyperinsulinemia as a result of insulin resistance [15], consumption of a high-fat diet negatively alters intestinal morphology [3], gut hormone secretion [16] and immune responses [17]. It is therefore imperative to understand the full consequences of kisspeptin on enteroendocrine and islet cells in obesity-diabetes.

Accordingly, this study aims to investigate the effects of kisspeptin-10 administration on both enteroendocrine and pancreatic islets cells in the context of diet-induced obesity. Specifically, we examine how kisspeptin-10 influences intestinal and pancreatic morphology, as well as gut hormone immunoreactivity, in female mice fed a high-fat diet (HFD). The naturally occurring form kisspeptin-10 was employed based on its GPR54-mediated bioequivalence to the full-length kisspeptin-54 peptide [18]. Female mice were selected due to the well-established interplay between kisspeptin and oestrogen in regulating the HPG axis, alongside heightened metabolic sensitivity observed in females [19]. By exploring the influence of kisspeptin-10 on both islet and intestinal endocrine cell populations in HFD mice, this study provides important insight into kisspeptin’s role in modulating the gut–pancreatic axis under conditions of metabolic stress, with relevance to obesity and its associated endocrine disturbances.

## 2. Methods

### 2.1. Animals

Female NIH Swiss mice (4–6 weeks old, Envigo, Huntingdon, UK) were housed individually with nesting in an air-conditioned room at 22 ± 2 °C with a 12 h light and dark cycle and ad libitum access to drinking water and standard rodent diet (10% fat, 30% protein and 60% carbohydrate; 12.99 kJ/g, Trouw Nutrition, Northwich, UK). At 7 weeks of age mice were singly caged and provided with a HFD (45% fat, 35% carbohydrate and 20% protein; 26.15 kJ/g, Special Diet Services, Witham, UK) for 16 weeks. Female NIH Swiss mice, an outbred strain, were selected for this study as they are susceptible to HFD-induced obesity, particularly after the post-pubertal period [20]. Following 16 weeks of high-fat feeding, HFD mice (*n* = 4–6) were then administered twice-daily intraperitoneal (i.p.) injections of saline vehicle (0.9% NaCl) or kisspeptin-10 (25 nmol/kg body weight, Synpeptide Co., Ltd., Shanghai, China, provided at greater than 95% purity) for the next 21 days at 09:00 and 17:00, with all ND mice also receiving twice-daily saline injections. Body weight and food intake were measured at regular intervals throughout the 21-day treatment regimen, and energy intake calculations based on the specific energy content, and quantity consumed, of each diet. The peptide dosing regimen was based on previous positive observations with related neuroendocrine peptides within a similar experimental system [21,22], and twice-daily injections owing to the relatively short half-life of kisspeptin-10 [6]. Kisspeptin has also been employed at comparable nmol/kg doses in earlier studies, further supporting the suitability for the current study [23,24]. While kisspeptin-54 represents the full-length physiologically active peptide, kisspeptin-10, the shortest biologically active fragment, was selected for the current study due to its improved blood–brain barrier penetration and full efficacy at the GPR54 receptor [5]. All experiments were approved by Ulster University Animal Welfare and Ethical Review Body (AWERB) (approved date of 20 January 2021) as well as under a UK Home Office Animal project licence number PPL2902 (approved date of 26 April 2021), conducted in accordance with the UK Animals (Scientific Procedures) Act 1986 and EU Directive 2010/63EU. Experiments were then reported in line with the ARRIVE (Animal Research: Reporting of In Vivo Experiments) guidelines.

### 2.2. Tissue Processing

Mice were euthanized by lethal inhalation of CO_2_ followed by cervical dislocation. Small intestine (ileum) and pancreatic tissues were excised from mice after 21 days treatment with kisspeptin-10 and fixed for 48 h in paraformaldehyde (4% *w*/*v* in phosphate-buffered saline (PBS)) to preserve cellular architecture. Tissues were then placed in an automated tissue processor, which involved dehydrating in 70% to 100% ethanol followed by xylene immersion to clear the tissue before paraffin embedding. Embedded tissues were then sectioned (5 μm thickness) and placed on poly-l-lysine coated slides [25].

### 2.3. Immunohistochemistry

To assess positive immunoreactivity for insulin, glucagon, PYY, GLP-1, GIP, Ki-67 and TUNEL, as appropriate, ileum and pancreatic sections were dewaxed in xylene for 30 min, before being rehydrated with decreasing concentrations of ethanol. Sections were blocked with 2.5% bovine serum albumin (BSA) and then incubated with designated primary antibody (Table 1) overnight. On day 2, sections were rinsed in PBS and incubated with appropriate secondary antibody (Alexa Fluor^®^ 594 for red and Alexa Fluor^®^ 488 for green, both Invitrogen, Paisley, UK; Table 1) for 1 h at 37 °C. After PBS wash, slides were then incubated with DAPI for 15 min at 37 °C [25]. Finally, sections were mounted on coverslips using antifade mounting media before being viewed at 40× magnification using an Olympus IX51 inverted microscope and photographed using a DP70 digital camera system (Tokyo, Japan).

**Table 1 biomolecules-15-01591-t001:** Target, host and source of primary and secondary antibodies employed for immunofluorescent imaging experiments.

Primary antibodies			
**Target**	**Host**	**Dilution**	**Source**
Insulin	Mouse	1:500	Abcam, ab6995
Glucagon	Guinea pig	1:200	Raised in-house PCA2/4
PYY	Rabbit	1:500	Abcam, ab22663
GLP-1	Rabbit	1:200	Raised in-house XJIC8
GIP	Rabbit	1:400	RIC34/111J, kindly donated by Professor L Morgan, Guildford, UK
Ki-67	Rabbit	1:200	Abcam, ab15580
**Secondary antibodies**			
**Host and target**	**Reactivity**	**Dilution**	**Fluorescent dilution and source**
Goat IgG	Mouse	1:500	Alexa Flour 594, Invitrogen, UK
Goat IgG	Guinea pig	1:500	Alexa Flour 488, Invitrogen, UK
Goat IgG	Rabbit	1:500	Alexa Flour 594, Invitrogen, UK

### 2.4. Image Analysis

Image J software (Version 1.54) was used to assess total ileal crypt depth and villi length from grayscale DAPI images using the straight-line function. The total number of GLP-1, GIP, and PYY immunopositive cells, along with the number of DAPI-stained nuclei, were quantified using QuPath (version 0.5.1) employing the positive cell detection function. Villi length was measured from the villus tip to the villus–crypt junction, and crypt depth was measured from the crypt base, using the straight-line tool in ImageJ. In addition, Cell^F^ software (Version 4) was used to analyse images to quantify islet area, beta- and alpha-cell areas and then PyCreas software (Version 1) to measure insulin and glucagon radius and relative islet cell areas [26]. As such, in murine islets with the classic mantle–core phenotype, the relative glucagon radius is expected to be larger than the relative insulin radius. Reductions in the relative glucagon radius indicate infiltration of alpha cells towards the islet centre, known to negatively impact rodent islet function [3]. Relative glucagon or insulin area was calculated from the ratio of glucagon or insulin cell pixel area, as appropriate, compared to total islet pixel area [27]. For beta-cell proliferation, insulin and Ki-67 co-positive cells were counted, whereas for apoptosis, insulin and TUNEL co-positive cells were counted, as described previously [25]. All measurements were performed following standardised protocols to ensure accuracy, reproducibility, and comparability across experimental groups [3,16].

### 2.5. Biochemical Analysis

Non-fasting blood glucose was directly measured from the cut tip on the tail vein of non-fasted conscious mice using a hand-held Ascencia Contour blood glucose metre (Bayer Healthcare, Newbury, Berkshire, UK). For terminal plasma analysis, blood was collected in chilled heparin/fluoride-coated microcentrifuge tubes (Sarstedt, Numbrecht, Germany) and centrifuged for 15 min at 1500× *g* using a Beckman microcentrifuge (Beckman Instruments, Galway, Ireland) to separate plasma. Commercially available assays were then used to assess circulating hormone levels including FSH, LH, progesterone and corticosterone (ligand assay and analysis core; Centre for Research in Reproduction, University of Virginia, VA, USA) as well as PYY (mouse PYY ELISA, ORB441862-BOR, Stratech Scientific, Cambridge, UK), GLP-1 (GLP-1 total ELISA, EZGLP-1T-36K, Millipore, Darmstadt, Germany) and GIP (rat/mouse GIP ELISA, EZRMGIP-55K, Millipore, Darmstadt, Germany), according to individual manufacturer’s instructions. For all terminal plasma analysis *n* = 4 was employed due to restrictions in blood volumes obtainable from mice.

### 2.6. Statistical Analysis

GraphPad PRISM (version 8.0) software was used to perform statistical analysis. Values are expressed as mean ± S.E.M. Comparative analyses between groups were conducted using two-way ANOVA or one-way ANOVA, using Bonferroni post hoc test as appropriate. Groups of data were considered to be significant if *p* < 0.05.

## 3. Results

### 3.1. Effects of HFD and Kisspeptin-10 Treatment on Body Weight, Glucose and Energy Intake

After 16 weeks on HFD, female mice were obese and mildly hyperglycaemic when compared with ND controls. Kisspeptin-10 significantly (*p* < 0.05–0.01) decreased body weight on days 7, 14 and 22 when compared to saline-treated HFD controls (Figure 1A), with non-fasting blood glucose also decreased (*p* < 0.05–0.01) on days 7, 14 and 22 (Figure 1B). In fact, by the end of the study body weight and blood glucose levels were almost identical in kisspeptin-10 treated HFD and healthy control ND mice, being 32.4 ± 0.9 vs. 35.1 ± 1.5 g and 7.2 ± 0.2 vs. 7.2 ± 0.3 mmol/L, respectively. Energy intake was significantly (*p* < 0.001) elevated in HFD mice, and kisspeptin-10 returned this parameter to ND control levels at all timepoints monitored (Figure 1C).

### 3.2. Effects of HFD and Kisspeptin-10 Treatment on Circulating Hormones

HFD increased (*p* < 0.05) plasma FSH levels, which were further augmented (*p* < 0.01) by kisspeptin-10 treatment (Figure 2A). Circulating LH demonstrated a similar pattern to FSH, with elevated (*p* < 0.05) concentrations in kisspeptin-10 treated HFD mice compared to ND mice (Figure 2B). There were no significant differences in circulating progesterone and corticosterone between the various groups (Figure 2C,D). In terms of gut hormone levels, plasma GIP was similar across all mice (Figure 2E). Whilst GLP-1 and PYY were not significantly altered in HFD mice, kisspeptin-10 intervention reduced (*p* < 0.05) the circulating GLP-1 and PYY concentrations when compared to saline-treated HFD controls (Figure 2F,G).

### 3.3. Effects of HFD and Kisspeptin-10 Treatment on Ileal Enteroendocrine Morphology and Cell Distribution

Ileal crypt depth was similar in ND and HFD mice but reduced (*p* < 0.05–0.001) by kisspeptin-10 treatment (Figure 3A). On the other hand, villi length was increased (*p* < 0.001) in HFD mice and was returned to normal levels by treatment with kisspeptin-10 (Figure 3B). Representative images of ileal tissues form the three different groups of mice are shown in Figure 3C. Further to this, immunofluorescence images of ileal sections stained positively for GIP, GLP-1 and PYY from each treatment group are shown in Figure 4A, Figure 5A and Figure 6A, respectively. There was no significant difference in the percentage of GIP, GLP-1 and PYY positively stained ileal cells between saline-treated HFD and ND mice (Figure 4B, Figure 5B and Figure 6B). However, kisspeptin-10 administration increased (*p* < 0.05) the percentage of GIP-positive ileal cells compared to saline-treated HFD mice (Figure 4B) and GLP-1 positive cells (*p* < 0.05) when compared to ND control mice (Figure 5B). However, changes in ileum cell composition were not observed when crypt and villus compartments were analysed separately (Figure 4C,D and Figure 5C,D). Finally, crypt and villi PYY immunoreactivity within the ileum was unaltered across all groups of mice (Figure 6A,C–E). Negative control ileal images incubated with secondary antibody only are also shown to confirm that observed signals are from the intended antibody reactions only (Figure 6B).

### 3.4. Effects of HFD and Kisspeptin-10 Treatment on Pancreatic Islet Morphology

Pancreatic islet morphology was assessed by quantifying islet, beta-cell and alpha-cell areas across treatment groups. No significant differences were observed in total islet, insulin-positive (beta cell) or glucagon-positive (alpha cell) areas between all groups of mice (Figure 7A–C). The numbers of islets per mm^2^ of pancreatic tissue were also comparable across groups (Figure 7D). In addition, the relative insulin radius remained unchanged across all groups (Figure 7E). However, the relative insulin–islet area was significantly reduced in HFD mice (*p* < 0.001), but restored to normal levels by kisspeptin-10 treatment (Figure 7F). Similarly, HFD feeding decreased (*p* < 0.05) the relative glucagon radius, suggesting alpha-cell displacement towards the islet centre, which was reversed by kisspeptin-10 (Figure 7G). Interestingly, kisspeptin-10 treatment also elevated (*p* < 0.01) the relative glucagon–islet area when compared to saline-treated HFD mice (Figure 7H). Representative images of pancreatic islets stained for insulin and glucagon from each group of mice are presented in Figure 7K, with negative control, secondary antibody only, images presented in Figure 7L.

### 3.5. Effects of HFD and Kisspeptin-10 Treatment on Pancreatic Beta-Cell Turnover

Quantitative analysis revealed that beta-cell proliferation and apoptosis frequencies remained unchanged in HFD mice when compared to ND controls (Figure 7I,J). However, kisspeptin-10 increased (*p* < 0.05) beta-cell proliferation frequency relative to HFD mice (Figure 7I) but had no impact on beta-cell apoptotic rates (Figure 7J).

## 4. Discussion

Prolonged consumption of a HFD can disrupt the HPG axis [27], with kisspeptin being a well-established stimulator of GnRH secretion leading to downstream pulsatile release of LH and FSH [28]. As such, in the current setting, HFD female mice presented with elevated FSH, which was further augmented by kisspeptin-10 treatment. Kisspeptin is generally considered to exert a more pronounced effect on LH [29], but circulating concentrations of LH and FSH are linked [30], in agreement with similar profiles for both hormones within each treatment group. In support of this, kisspeptin administration was previously noted to elevate both FSH and LH in female rats [4], confirming a central impact of peripherally administered kisspeptin. Thus, similar to the parent peptide, kisspeptin-10 can modulate reproductive function in both rodents [31] and humans [32]. Although not the primary focus of the current study, it is clear that GRP54 signalling is profoundly involved in reproductive neuroendocrine regulation and could represent a viable target for reversing the negative impact of obesity on fertility [33]. Indeed, excess adiposity is often associated with menstrual irregularities and conditions such as polycystic ovary syndrome (PCOS), characterised by altered LH/FSH profiles and metabolic dysfunction [27,34].

In keeping with this and in good harmony with the current findings, earlier work demonstrated that kisspeptin signalling plays a role in energy homeostasis and body weight regulation, especially in females [35,36]. Fittingly, sustained intraperitoneal administration of kisspeptin-10 in female HFD mice reduced body weight, non-fasting glucose and energy intake. Similar findings have been reported with chronic central kisspeptin-10 administration [37], suggesting both central and peripheral mechanisms to be involved [33]. While centrally administered kisspeptin is known to influence hypothalamic hunger/satiety pathways such as NPY/AgRP and POMC/CART neurons [38], potential mechanisms beyond the brain warrant further investigation, particularly since GPR54 expression is evidenced in various metabolically active tissues such as the gut and endocrine pancreas [5,39]. Notably, GPR54 expression is evidenced in both glucagon-secreting alpha cells and the insulin-secreting beta cells, with little or no expression in exocrine pancreatic tissue [39]. Whilst in the gut, GPR54 is detectable in the small intestine and seems to be predominantly within the enteroendocrine cell population [33]. Moreover, given that kisspeptin-10 therapy essentially normalised body weight in our HFD model, it would be interesting to assess the impact of weight loss alone on ileal cell and pancreatic islet morphology possibly using pair-feeding or calorie restriction, in order to determine direct effects of sustained GPR54 activation.

To our knowledge, only one study has previously investigated the impact of kisspeptin on gastrointestinal physiology, demonstrating enhanced gut transit following central (ICV) administration of kisspeptin-13 [9]. Since a HFD is known to slow intestinal transit [40], alongside the abundance of PYY and GLP-1-secreting ileal L-cells [41], we examined whether persistent kisspeptin-10 treatment could modulate ileal structure and hormone expression. As clearly identified in the current study, a HFD typically increases villus length in adaptation to enhanced nutrient absorption [42]. Kisspeptin-10 fully reversed this alteration in ileal morphology, which could be linked to reduced appetite in these mice. Furthermore, ileal crypt depth was decreased by kisspeptin-10, implying reduced enteroendocrine cell turnover [43]. Further studies are required to assess these effects in relation to nutrient absorption and turnover of intestinal enterocytes and endocrine cells. Notably, however, kisspeptin-10 intervention augmented the overall number of GIP and GLP-1 ileal positive cells, with both hormones recently attracting attention in respect to benefits for obesity, diabetes and various other metabolic conditions [44].

Given the experimental model employed alongside the impact of both GIP and GLP-1 on lipid metabolism [45,46], as well as evidence that kisspeptin can modulate lipogenesis and lipolysis [47], changes in enteroendocrine cell expression in HFD mice may reflect gut-related adaptations that are important for the beneficial effects of kisspeptin-10 on overall metabolism. That said, alterations in cell immunoreactivity did not correspond with changes circulating gut hormone levels, with kisspeptin-10 reducing plasma GLP-1 and PYY. Further investigation is required, but HFD and concurrent kisspeptin treatment may affect gut hormone secretion rather than cellular abundance or possibly impact clearance of these hormones. However, there is a delicate balance between hormone synthesis and secretion [48] that is also likely to be important in this context. Indeed, it is plausible that observed effects on gut hormones are upstream of kisspeptin signalling, rather than being directly regulated by GPR54 activity [49]. As such, further studies are needed to fully delineate the bi-directional relationship between kisspeptin signalling and gut hormone synthesis and secretory dynamics.

A close link exists between the gut and endocrine pancreas, often referred to as to gut–pancreas axis [13]. Owing to the protective effects of oestrogen [50], female HFD-fed mice tend to exhibit less significant pancreatic islet alterations than male counterparts [20,51], which seemed to be the case in the current study. However, exploiting a more in-depth investigation of the localisation and distribution of endocrine cell types within the islets [28] confirmed that HFD promoted alpha-cell centralisation as well as reducing islet insulin expression in female mice [3]. More notably, kisspeptin-10 treatment protected against HFD-associated islet architectural abnormalities, alongside increasing relative alpha-cell area. Since morphometric analysis was the primary endpoint of the current study, we did not assess circulating islet hormone levels, but our approach still provides potential insights into kisspeptin-10-mediated effects on insulin and glucagon secretion and function, as evidenced in related preclinical studies [52].

In addition to this, although the mechanism behind this increase in alpha-cell area is less well understood, the ability of kisspeptin-10 to reverse the negative impact of high-fat feeding on islet morphology could be linked to beneficial effects on islet cell plasticity, which have recently been shown to be fundamental for maintaining beta-cell identity and mass [53]. Thus, alpha cells are recognised a key progenitor cell type for beta-cell neogenesis [54]. Moreover, pregnancy-induced expansion of beta-cell mass has recently been suggested to involve changes in islet cell transdifferentiation events [55], with kisspeptin already shown to play a physiological role in the islet adaptation to pregnancy in humans [56]. Furthermore, glucagon has also been demonstrated to regulate peripheral GPR54 expression [9], that may also influence kisspeptin-10 driven changes in alpha-cell mass. Kisspeptin-mediated support of beta-cell mass is further strengthened through observations of enhanced proliferative capacity of these cells in kisspeptin-10-treated HFD mice. Interestingly, kisspeptin has also been suggested to play a role in beta-cell autophagy in HFD rodents, protecting against glucolipotoxicity-induced apoptosis [10], corresponding well with our findings on beta-cell turnover. These data support the idea that peripheral kisspeptin-10 confers structural protection to islets in obesity, which merits further consideration for diseases of beta-cell loss such as diabetes [57]. Howbeit, equivalent studies in male mice would still be required to uncover whether the observed effects of kisspeptin-10 are specific to female mice. Indeed, it might be expected to uncover a certain degree of sexual dimorphism, given the intimate relationship between kisspeptin and oestrogen in terms of metabolic control [6]. Finally, although there is good evidence for GPR54 expression on islet cells [39], we are unable to determine the extent to which effects on alpha- and beta-cells are direct, or secondary to systemic metabolic alterations induced by sustained kisspeptin-10 administration.

## 5. Conclusions

The current study provides novel insights into the impact of peripherally administered kisspeptin-10 to modulate gut morphology, enteroendocrine cell makeup as well as pancreatic islet cell localisation and distribution in response to high-fat feeding in female mice. The work represents initial evidence that kisspeptin-10 simultaneously modulates gut hormone-producing cell populations and protects pancreatic islet architecture in female HFD-fed mice, supporting a potential therapeutic role for kisspeptin in obesity-related insulin resistance and fertility disorders.

## Figures and Tables

**Figure 1 biomolecules-15-01591-f001:**
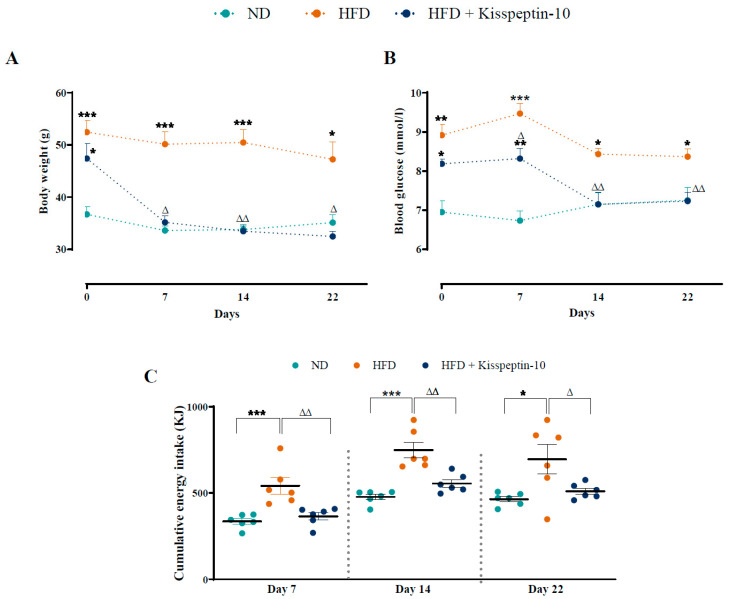
Effects of kisspeptin-10 on (**A**) body weight, (**B**) blood glucose and (**C**) energy intake in female HFD mice. Parameters were measured during twice-daily treatment with kisspeptin-10 (25 nmol/kg bw) for 21 days in HFD female mice. Values are mean ± SEM (*n* = 6). * *p* < 0.05, ** *p* < 0.01 and *** *p* < 0.001 compared to ND control mice. ^Δ^ *p* < 0.05, ^ΔΔ^ *p* < 0.01 compared to HFD mice.

**Figure 2 biomolecules-15-01591-f002:**
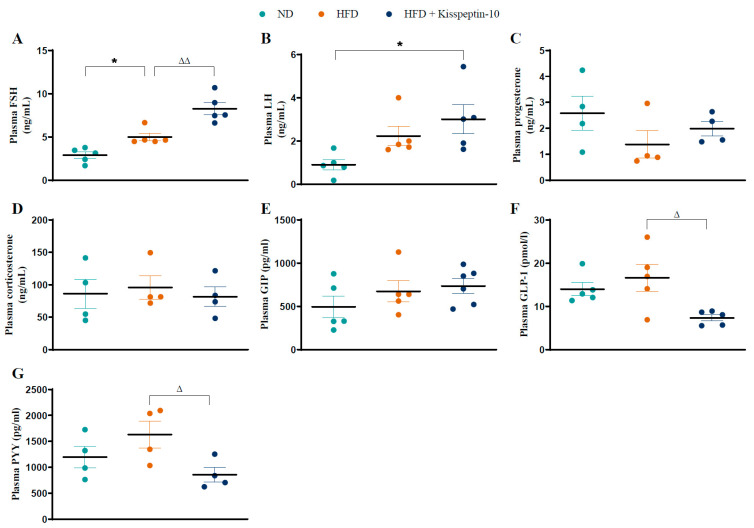
Effects of kisspeptin-10 on circulating hormone levels in female HFD mice. Parameters were measured after 21 days of twice-daily treatment with kisspeptin-10 (25 nmol/kg bw) in HFD mice. Plasma (**A**) FSH, (**B**) LH, (**C**) progesterone, (**D**) corticosterone, (**E**) GIP, (**F**) GLP-1 and (**G**) PYY. Values are mean ± SEM (*n* = 4). * *p* < 0.05 compared to ND control mice. ^Δ^
*p* < 0.05 and ^ΔΔ^
*p* < 0.01 compared to HFD mice.

**Figure 3 biomolecules-15-01591-f003:**
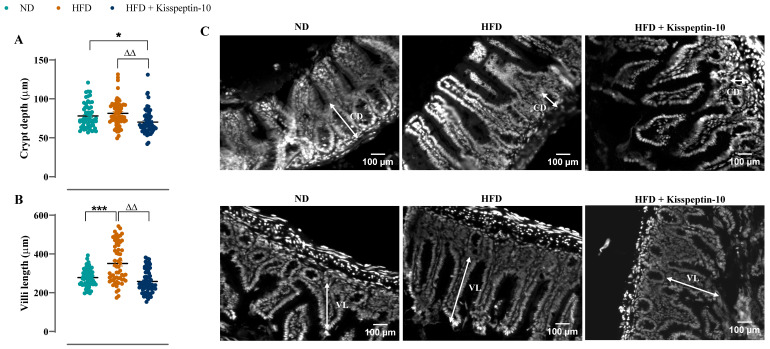
Effects of kisspeptin-10 on ileum morphology in female HFD mice. Parameters were measured after 21 days of twice-daily treatment with kisspeptin-10 (25 nmol/kg bw) in HFD mice. (**A**) Crypt depth and (**B**) villi length and (**C**) representative images of ileum where CD is crypt depth and VL is villi length. 60 crypts and villi were analysed from each group. Values are mean ± SEM (*n* = 6). * *p* < 0.05 and *** *p* < 0.001 compared to ND control mice. ^ΔΔ^ *p* < 0.01 compared to HFD mice.

**Figure 4 biomolecules-15-01591-f004:**
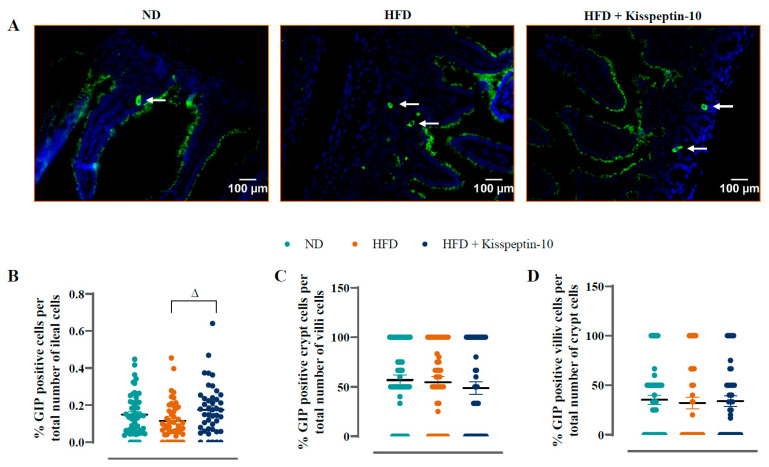
Effects of kisspeptin-10 on ileal GIP cell distribution in female HFD mice. Parameters were measured after 21 days of twice-daily treatment with kisspeptin-10 (25 nmol/kg bw) in HFD mice. (**A**) Representative images of ileum stained for GIP (green) and DAPI (blue). Related quantification of (**B**) % GIP positive cells per total number of ileal cells, (**C**) % GIP positive cells per total number of crypt cells and (**D**) % GIP positive cells per total number of villi cells. A total of 60–70 images were analysed per group. White arrows indicate positively stained cells. Values are mean ± SEM (*n* = 6). ^Δ^
*p* < 0.05 compared to HFD mice.

**Figure 5 biomolecules-15-01591-f005:**
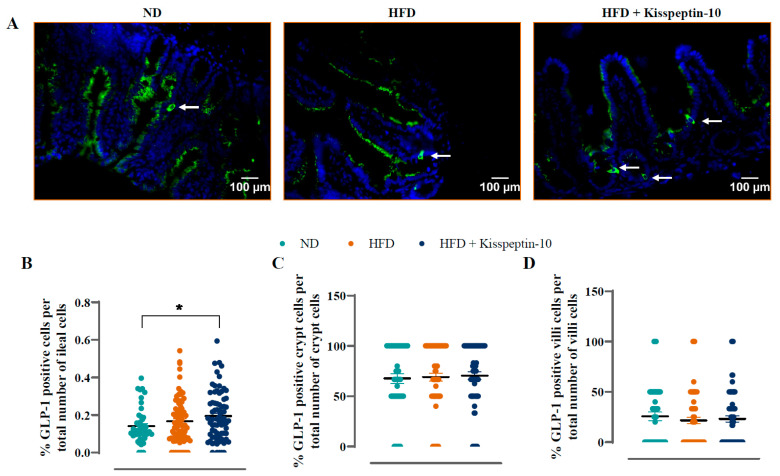
Effects of kisspeptin-10 on ileal GLP-1 cell distribution in female HFD mice. Parameters were measured after 21 days of twice-daily treatment with kisspeptin-10 (25 nmol/kg bw) in HFD mice. (**A**) Representative images of ileum stained for GLP-1 (green) and DAPI (blue). Related quantification of (**B**) % GLP-1 positive cells per total number of ileal cells, (**C**) % GLP-1 positive cells per total number of crypt cells and (**D**) % GLP-1 positive cells per total number of villi cells. A total of 60–70 images were analysed per group. White arrows indicate positively stained cells. Values are mean ± SEM (*n* = 6). * *p* < 0.05 compared to ND control mice.

**Figure 6 biomolecules-15-01591-f006:**
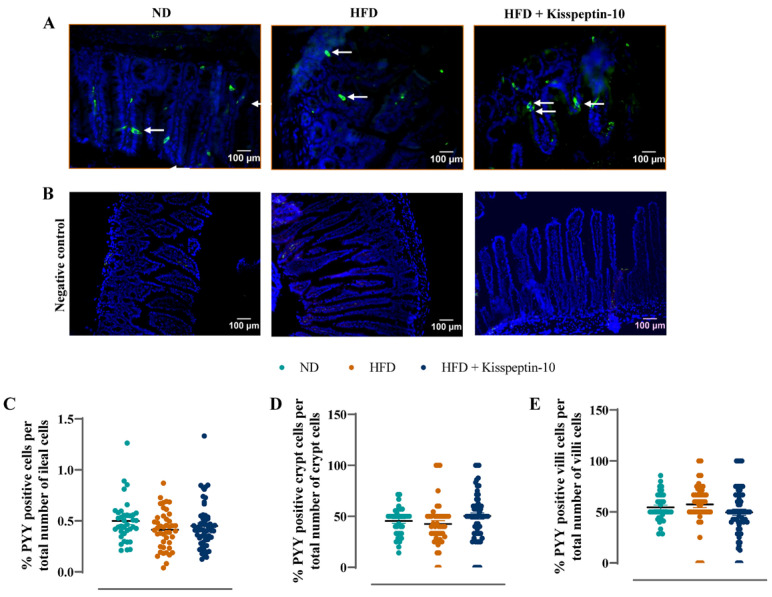
Effects of kisspeptin-10 on ileal PYY cell distribution in female HFD mice. Parameters were measured after 21 days of twice-daily treatment with kisspeptin-10 (25 nmol/kg bw) in HFD mice. (**A**) Representative images of ileum stained for PYY (green) and DAPI (blue) as well as (**B**) negative control images with secondary antibody incubation only. Related quantification of (**C**) % PYY positive cells per total number of ileal cells, (**D**) % PYY positive cells per total number of crypt cells and (**E**) % PYY positive cells per total number of villi cells. A total of 60–70 images were analysed per group. White arrows indicate positively stained cells. Values are mean ± SEM (*n* = 6).

**Figure 7 biomolecules-15-01591-f007:**
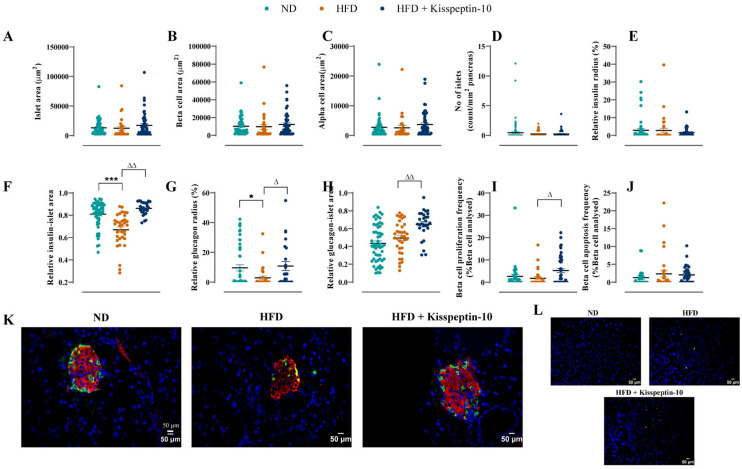
Effects of kisspeptin-10 on pancreatic islet architecture, morphology, beta-cell growth and survival in female HFD mice. Parameters were measured after 21 days of twice-daily treatment with kisspeptin-10 (25 nmol/kg bw) in HFD mice. (**A**) Islet area, (**B**) beta-cell area, (**C**) alpha-cell area, (**D**) number of islets per mm^2^ pancreas, (**E**) relative insulin radius, (**F**) relative insulin-islet area, (**G**) relative glucagon radius, (**H**) relative glucagon-islet area, (**I**) beta-cell proliferation, (**J**) beta-cell apoptosis, (**K**) representative images of islets stained for insulin (red), glucagon (green) and DAPI (blue) as well as (**L**) negative control images with secondary antibody incubation only. A total of 50–60 islets were analysed from each group. Values are mean ± SEM (*n* = 6). * *p* < 0.05 and *** *p* < 0.001 compared to ND control mice. ^Δ^
*p* < 0.05 and ^ΔΔ^
*p* < 0.01 compared to HFD mice.

## Data Availability

The authors declare that the data supporting the findings of this study are available within the article. Any additional raw data supporting the conclusions of this article will be made available by the corresponding author, without undue reservation.

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
