# Peer review of "Kisspeptin-10 Ameliorates Obesity-Diabetes with Diverse Effects on Ileal Enteroendocrine Cells and Pancreatic Islet Morphology in High-Fat Fed Female Mice"

_biomolecules, 2025, doi:10.3390/biom15111591_

Round 1

Reviewer 1 Report

Comments and Suggestions for Authors

This is a well-written paper that reports the outcomes from a simple experiment investigating the therapeutic potential of kisspeptin-10. One concern is the small size of the study (n’s of 4 to 6). However, there are significant results so this is perhaps not a major concern.

The authors report a study in which administration of kisspeptin-10 reverses diet-induced obesity in female NIH Swiss mice. The evidence linking kisspeptin to body weight regulation is inconsistent. Genetic models suggest loss of the protein can cause weight gain, however the responses to exogenous peptide have not been thoroughly investigated. There are reports of kisspeptin reducing food intake, although there is some inconsistency in the literature.

The current findings suggest that, at least in females, kisspeptin administration can reduce food intake and body weight.

There are some interesting effects on gut peptides (PYY, GLP-1) which suggest this peptide interacts with these critical pathways. Mostly, the data are consistent with the observed effects on energy balance and support a role for this peptide in body weight regulation.

Questions for the authors:

  • The authors don’t discuss the distribution of GPR54 expression in the gut. Can more information be provided on the cell type expression of this receptor in the intestinal tract and pancreas?
  • A closer examination of how reversing diet-induced obesity affects the systems studied is needed. Specifically, the authors should investigate whether weight loss has similar effects – it may be that the responses observed are secondary to weight loss. This should be discussed.
  • Would the authors consider the outcomes to be specific for females, given the sexual dimorphism observed on the impact of modulating this pathway?

Author Response

Reviewer 1:

We thank the Reviewer for their evaluation of our manuscript and in particular the comment that this is a ‘well-written paper’. As the Reviewer states, the impact of kisspeptin to body weight regulation and related factors is currently not well described in the published literature. We designed our study to fill this knowledge gap using minimum numbers of animals, whilst still maintaining sufficient statistical power to observe obvious changes, directly in line with the 3R principles. Our responses to the specific queries raised by the Reviewer are then considered below.

Specific comments

  1. The Reviewer comments ‘The authors don’t discuss the distribution of GPR54 expression in the gut. Can more information be provided on the cell type expression of this receptor in the intestinal tract and pancreas?’. The authors thank the Reviewer for the opportunity to expand on this important matter. As noted in the Discussion section of the original manuscript, GPR54 is expressed in both the endocrine pancreas and the gut (see lines 248-249). We are happy to provide additional information in this regard. We have therefore added the following text to the Discussion section of the revised manuscript, that reads as follows (lines 249-252): Notably, GPR54 expression is evidenced in both glucagon-secreting alpha-cells and the insulin-secreting beta cells, with little or no expression in exocrine pancreatic tissue.39 Whilst in the gut, GPR54 is detectable in the small intestine and seems to be predominantly within the enteroendocrine cell population.33’.

  1. The Reviewer remarks ‘A closer examination of how reversing diet-induced obesity affects the systems studied is needed. Specifically, the authors should investigate whether weight loss has similar effects – it may be that the responses observed are secondary to weight loss. This should be discussed’. The authors thank the Reviewer for this comment and agree that knowledge of the impact of weight loss on ileal enteroendocrine cells and pancreatic islet morphology would be of interest. We are unable to conduct such experiments since the animals have now been culled and fear that interpretation of pair-feeding data is complicated by encouraging animals to unnaturally meal-feed at particular times of the day. However, we are keen to add some further discussion on this point, as suggested. To highlight this in the revised manuscript, we have added the following sentence to the Discussion section (lines 252-256), that reads as follows: ‘Moreover, given that kisspeptin-10 therapy essentially normalized body weight in our HFD model, it would be interesting to assess the impact of weight loss alone on ileal cell and pancreatic islet morphology possibly using pair-feeding or calorie restriction, in order to determine direct effects of sustained GPR54 activation’.

  1. The Reviewer enquires ‘Would the authors consider the outcomes to be specific for females, given the sexual dimorphism observed on the impact of modulating this pathway?’. The Reviewer raises an interesting point. As noted in our manuscript, we chose to investigate effects of kisspeptin-10 on ileal enteroendocrine cell and pancreatic islet morphology in female HFD mice because of the well-established interplay between kisspeptin and oestrogen in regulating the hypothalamic-pituitary-gonadal axis, alongside their susceptibility to HFD-induced. However, we agree that sexual dimorphism can not be ruled out. To reinforce this within the revised manuscript, the following text has been added to the Discussion section (lines 313-317), that reads as follows: ‘Howbeit, equivalent studies in male mice would still be required to uncover whether the observed effects of kisspeptin-10 are specific to female mice. Indeed, it might be expected to uncover a certain degree of sexual dimorphism, given the intimate relationship between kisspeptin and oestrogen in terms of metabolic control.6’.

Reviewer 2 Report

Comments and Suggestions for Authors

This manuscript explores the effects of kisspeptin-10 administration on intestinal and pancreatic morphology and hormone expression in female mice subjected to a high-fat diet (HFD). The study addresses an interesting topic, as kisspeptin signaling has emerging roles in metabolism and energy balance beyond its established reproductive function. Investigating its impact on the gut-pancreas axis in the context of obesity is scientifically relevant and potentially novel.

The study design appears logical, and the results could contribute to understanding the metabolic actions of kisspeptin. However, several aspects of the experimental design, data validation, and presentation require clarification or additional supporting evidence before the manuscript can be considered for publication. Below are the comments that need to be addressed:

  1. In the method section, it is not clearly mentioned whether the mice were fed HFD for 16 weeks and additional 3 weeks on HFD + kisspeptin injections or 13 weeks on HFD and 3 weeks on kisspeptin injections.
  2. The authors report cumulative energy intake in Figure 1C and mention that mice were single-housed, which enables accurate individual measurements. However, the methodology for quantifying energy intake is not described. It would be helpful if the authors specify how energy intake (kJ) was calculated from the amount of food consumed. Additionally, was there any difference in energy intake before the injection between the groups?
  3. At what point were mice single housed? Are they single housed right after the weaning? Single housing induces stress to mice and causes body weight loss. Mice need to acclimatize at least for a week before starting the experiment.
  4. The authors mentioned that HFD mice (n = 4–6) were administered twice-daily intraperitoneal (i.p.) injections of either saline vehicle (0.9% NaCl) or kisspeptin-10. What is the reason for giving two injections per day? Is it due to the short half-life of the peptide? What was the time interval between the two injections? Additionally, were the ND mice also injected with vehicle?
  5. In the figure 1A legend it is mentioned that the mice numbers per group were between 4 to 6. But in Figure 1C, there are at least 5 mice per group.
  6. In Figure 1A, body weight appears to decrease by approximately 10 g after one week of Kisspeptin-10 administration, with a modest reduction persisting during the second and third weeks. It would be more informative if the authors provided daily body-weight measurements rather than weekly averages, as this would allow a clearer understanding of the temporal pattern and magnitude of weight change following treatment.
  7. The author has not mentioned the euthanasia method. Was fasting performed before euthanasia? How was plasma collected?
  8. The images of Fig3C are not clear enough. Additionally, it is not mentioned what kind of staining were done to obtain these images. HE staining can be considered.
  9. While in Fig 3C the authors have shown that HFD increases villus length and peptide treatment reverses this effect, it would strengthen the manuscript if the authors could provide complementary evidence. For instance, PAS staining or Ki67 immunostaining to confirm villus structure and proliferation, or functional assays of intestinal absorption could provide additional validation.
  10. The authors should include negative control images for all immunofluorescence experiments. This could include staining without primary antibody or using isotype controls to confirm the specificity of the signal.
  11. The quantification of pancreatic morphology and hormone immunoreactivity is clearly described and well presented. The authors could further enhance this section by briefly discussing whether the observed islet changes likely reflect direct kisspeptin-10 action or are secondary to systemic metabolic alterations.

Author Response

Reviewer 2:

We thank the Reviewer for their appraisal of our manuscript, and especially his/her comments that ‘The study addresses an interesting topic, as kisspeptin signaling has emerging roles in metabolism and energy balance beyond its established reproductive function’ and ‘The study design appears logical, and the results could contribute to understanding the metabolic actions of kisspeptin’. Our responses to the specific queries raised by the Reviewer are considered below.

Specific comments

  1. The Reviewer comments ‘In the method section, it is not clearly mentioned whether the mice were fed HFD for 16 weeks and additional 3 weeks on HFD + kisspeptin injections or 13 weeks on HFD and 3 weeks on kisspeptin injections. The authors apologise for any misunderstanding in terms of our study design. We can confirm that mice were maintained on a HFD for 16 weeks and kisspeptin-10 therapy then initiated for an additional 3 week period. To make this clearer in the revised manuscript, we have rewritten the relevant sentence in the Methods section to now read (lines 88-92): ‘Following 16 weeks of high fat feeding, HFD mice (n=6) were then administered twice daily intraperitoneal (i.p.) injections of saline vehicle (0.9% NaCl) or kisspeptin-10 (25 nmol/kg body weight, Synpeptide Co. Ltd., Shanghai, provided at greater than 95% purity) for the next 21 days at 09:00 and 17:00, with all ND mice also receiving twice daily saline injections’.

  1. The Reviewer notes ‘The authors report cumulative energy intake in Figure 1C and mention that mice were single-housed, which enables accurate individual measurements. However, the methodology for quantifying energy intake is not described. It would be helpful if the authors specify how energy intake (kJ) was calculated from the amount of food consumed. Additionally, was there any difference in energy intake before the injection between the groups?’. We thank the Reviewer for the opportunity to clarify this matter. Energy intake was calculated based on the energy content of the two diets employed, namely standard rodent maintenance diet and the 45% high fat diet, alongside amount of food consumed. We note that energy content of the standard maintenance diet was not provided in the original manuscript, and we apologise for this omission. The energy content of this diet, 12.99 kJ/g, is now provided within the revised text (please see line 84). We have also added an additional sentence to the Methods section (lines 92-94) to make this process clearer, which reads as follows: ‘Body weight and food intake were measured at regular intervals throughout the 21-day treatment regimen, with energy intake calculations based on the specific energy content, and quantity consumed, of each diet’. The authors can confirm that there was no difference in energy intake between the groups of HFD mice before the injection regimen began. We have chosen not to comment further on this within the revised manuscript to avoid any potential reader confusion.

  1. The Reviewer asks ‘At what point were mice single housed? Are they single housed right after the weaning? Single housing induces stress to mice and causes body weight loss. Mice need to acclimatize at least for a week before starting the experiment’. The authors can confirm that all mice were singly housed to facilitate measurement of cumulative food intake. Single housing was implemented at the point of introduction of the high fat diet. Thus, all mice were in single cages for 16 weeks prior to commencement of peptide injections. To make this clearer in the revised text, we have reworded the following sentence in the Methods section, that now reads (lines 84-86): ‘At 7 weeks of age mice were singly caged and provided with a HFD (45% fat, 35% carbohydrate and 20% protein; 26.15 kJ/g, Special Diet Services, UK) for 16 weeks’.

  1. The Reviewer comments ‘The authors mentioned that HFD mice (n = 4–6) were administered twice-daily intraperitoneal (i.p.) injections of either saline vehicle (0.9% NaCl) or kisspeptin-10. What is the reason for giving two injections per day? Is it due to the short half-life of the peptide? What was the time interval between the two injections? Additionally, were the ND mice also injected with vehicle?’. The authors can confirm that use of two injections per day was related to the relatively short half-life of native kisspeptin-10. Injections were administered at 09:00 and 17:00 each day, and we can verify that all ND and HFD control mice received twice daily saline injections. This information has now been added to the Methods section (lines 88-97) and reads as follows: ‘Following 16 weeks of high fat feeding, HFD mice (n=4-6) were then administered twice daily intraperitoneal (i.p.) injections of saline vehicle (0.9% NaCl) or kisspeptin-10 (25 nmol/kg body weight, Synpeptide Co. Ltd., Shanghai, provided at greater than 95% purity) for the next 21 days at 09:00 and 17:00, with all ND mice also receiving twice daily saline injections. Body weight and food intake were measured at regular intervals throughout the 21-day treatment regimen, and energy intake calculations based on the specific energy content, and quantity consumed, of each diet. The peptide dosing regimen was based on previous positive observations with related neuroendocrine peptides within a similar experimental system,21,22 and twice daily injection owing to the relatively short half-life of kisspeptin-10.6’. We thank the Reviewer for this opportunity to improve the transparency of our experimental regimen.

  1. The Reviewer notes ‘In the figure 1A legend it is mentioned that the mice numbers per group were between 4 to 6. But in Figure 1C, there are at least 5 mice per group’. The Reviewer is correct in that Figure 1C displays data for 6 mice per treatment group. For all other subsequent quantitative analyses we also employed n=6, expect for data contained within Figure 2. We have now updated the Methods and Figure legend sections to make this clearer. We were restricted to n=4 for these plasma analyses due to limitations blood volume that we could obtain from the mice. A comment to this effect has now been added to the Methods section (lines 157-158) and reads as follows: ‘For all terminal plasma analysis n=4 was employed due to restrictions in blood volumes obtainable from mice’.

  1. The Reviewer comments ‘In Figure 1A, body weight appears to decrease by approximately 10 g after one week of Kisspeptin-10 administration, with a modest reduction persisting during the second and third weeks. It would be more informative if the authors provided daily body-weight measurements rather than weekly averages, as this would allow a clearer understanding of the temporal pattern and magnitude of weight change following treatment’. We thank the Reviewer for this comment. However, given the increased stress response from handling mice and administering twice daily injections, we choose to not to add to this by weighing mice on a daily basis. We therefore weighed mice only once weekly to reduce stress. Whilst we agree that knowledge of daily body weights might be of some interest, the critical information for body weight relates to the day of animal culling, which is provided. This being the timepoint when all subsequent quantitative analyses were conducted. To avoid any potential Reader confusion we have decided not to comment further on this matter within the revised manuscript.

  1. The Reviewer states ‘The author has not mentioned the euthanasia method. Was fasting performed before euthanasia? How was plasma collected?. The authors apologise for this oversight, and we have now added this information to the revised text. As such, details regarding the euthanasia method are available within the Methods section on line 109: ‘Mice were euthanized by lethal inhalation of CO2 followed by cervical dislocation’, with details of plasma collection also now within the Methods section on lines 148-151: ‘For terminal plasma analysis, blood was collected in chilled heparin/fluoride-coated micro-centrifuge tubes (Sarstedt, Numbrecht, Germany) and centrifuged for 15 min at 1500 g using a Beckman microcentrifuge (Beckman Instruments, Galway, Ireland) to separate plasma’. We thank the Reviewer for this chance to improve the quality of our manuscript.

  1. The Reviewer remarks ‘The images of Fig3C are not clear enough. Additionally, it is not mentioned what kind of staining were done to obtain these images. HE staining can be considered’. We are sorry that the images within Figure 3 are not clear enough for the Reviewer. As suggested, we have now updated these images and also added details to the Methods section in relation to the type of staining employed, which was DAPI staining in grayscale. The new information is available within lines 128-129 of the revised Methods section, and reads as follows: ‘Image J software was used to assess total ileal crypt depth and villi length from grayscale DAPI images using the straight-line function’.  

  1. The Reviewer comments ‘While in Fig 3C the authors have shown that HFD increases villus length and peptide treatment reverses this effect, it would strengthen the manuscript if the authors could provide complementary evidence. For instance, PAS staining or Ki67 immunostaining to confirm villus structure and proliferation, or functional assays of intestinal absorption could provide additional validation’. The authors thank the Reviewer for this comment, and this is something we had also contemplated. However, we considered direct measurement of villus length to provide solid, direct and well accepted evidence of intestinal growth. Unfortunately, functional assays are not possible given that all animals have now been culled. We had then considered intestinal proliferation staining with Ki-67, as we have already employed within the endocrine pancreas. However, unlike the endocrine pancreas where individual cell designation is possible using appropriate co-staining, the heterogeneous nature of intestinal cells, many of which are known to secrete various hormones and products, means this is not possible. We therefore decided not to follow up on this approach to avoid ambiguity in terms of cell type. We trust that the Reviewer can understand our decision making process in this regard. That said, to acknowledge the worthy comment raised by the Reviewer, the authors are happy to add the following text to the Discussion section to note that such additional studies would be of merit. This reads as follows (lines 266-267): ‘Further studies are required to assess these effects in relation to nutrient absorption and turnover of intestinal enterocytes and endocrine cells’.

  1. The Reviewer suggests ‘The authors should include negative control images for all immunofluorescence experiments. This could include staining without primary antibody or using isotype controls to confirm the specificity of the signal’. The authors thank the Reviewer for this comment, and we are happy to comply with their suggestion. We have now included appropriate negative control images for all immunofluorescence experiments, and updated the manuscript accordingly. Specifically, ileal tissue was incubated with appropriate secondary antibody only (Alexa Fluor® 488) and these images now added to Figure 6, panel B. For pancreatic islets, we have completed similar staining with secondary antibody only (both Alexa Fluor® 594 and Alexa Fluor® 488) and included these images within Figure 7, panel L. We have updated the Results section (lines 200-202 as well as lines 216-217) and figure legend sections accordingly. We are grateful for this chance to increase the transparency of our work.

  1. The Reviewer comments ‘The quantification of pancreatic morphology and hormone immunoreactivity is clearly described and well presented. The authors could further enhance this section by briefly discussing whether the observed islet changes likely reflect direct kisspeptin-10 action or are secondary to systemic metabolic alterations’. The authors thank the Reviewer for their appreciation of our pancreatic islet morphology and hormone immunoreactivity data. He/she then raises an excellent point in terms of whether these effects are directly related to GPR54 activation within islets, or because of systemic metabolic alterations induced by sustained kisspeptin-10 administration. We are unable to answer this question, but predict that it is likely a combination of both factors. Yet, without any evidence, we are reluctant to make any speculation in this regard. However, to more candidly address the Reviewers point, we have added the following sentence to the Discussion section to highlight this matter, that reads as follows (lines 317-320): ‘Finally, although there is good evidence for GPR54 expression on islet cells,39 we are unable to determine the extent to which effects on alpha- and beta-cells are direct, or secondary to systemic metabolic alterations induced by sustained kisspeptin-10 administration’.

Round 2

Reviewer 2 Report

Comments and Suggestions for Authors

Thank you for responding to all the comments.